# Pruning Points Detection of Sweet Pepper Plants Using 3D Point Clouds and Semantic Segmentation Neural Network

**DOI:** 10.3390/s23084040

**Published:** 2023-04-17

**Authors:** Truong Thi Huong Giang, Young-Jae Ryoo

**Affiliations:** 1Department of Electrical Engineering, Mokpo National University, Muan 58554, Jeonnam, Republic of Korea; tthgiang@ttn.edu.vn; 2Department of Electrical and Control Engineering, Mokpo National University, Muan 58554, Jeonnam, Republic of Korea

**Keywords:** 3D point cloud, pruning point, sweet pepper, semantic segmentation neural network

## Abstract

Automation in agriculture can save labor and raise productivity. Our research aims to have robots prune sweet pepper plants automatically in smart farms. In previous research, we studied detecting plant parts by a semantic segmentation neural network. Additionally, in this research, we detect the pruning points of leaves in 3D space by using 3D point clouds. Robot arms can move to these positions and cut the leaves. We proposed a method to create 3D point clouds of sweet peppers by applying semantic segmentation neural networks, the ICP algorithm, and ORB-SLAM3, a visual SLAM application with a LiDAR camera. This 3D point cloud consists of plant parts that have been recognized by the neural network. We also present a method to detect the leaf pruning points in 2D images and 3D space by using 3D point clouds. Furthermore, the PCL library was used to visualize the 3D point clouds and the pruning points. Many experiments are conducted to show the method’s stability and correctness.

## 1. Introduction

Sweet pepper is one of the most popular fruits for people today because of its nutrition, such as vitamins A, B, and C, and the minerals Ca, P, K, and Fe [1]. The amounts of these nutrients change at different maturity stages. For example, there is an increase in vitamin C during ripening [2]. They are also affected by growing conditions such as light, irrigation, and fertilizers [1,3]. In addition, the pruning technique plays an important role. Different strategies in pruning cause different fruit sizes, quality traits, and yield [4]. None-pruned plants could have more diseases than pruned plants [5]. Sweet pepper pruning includes shoot pruning and leaf pruning. Leaf pruning helps control aphids in sweet pepper crops [6]. Therefore, the pruning action is performed daily as a necessary farm activity.

Plant pruning is an easy task for humans, but it takes time. In contrast, it is a difficult task for machines. It must perform tasks such as recognizing the plant parts through camera sensors, detecting removable objects and the position of pruning points, and then controlling the robot arm to remove them. However, machines can work day and night continuously. It could save lots of time. Furthermore, it can reduce the risk of disease infection from laborers who usually get on the farm from outside. Automation plays an essential part in intelligent agriculture. This prompted us to study this subject and apply it to sweet pepper, one of the popular plants in smart farms. Leaf pruning was selected as this research object.

As with the jobs listed above, our research includes many phases. The first phase is recognizing the plant parts, which was solved by our previous study of the semantic segmentation neural network [7]. This neural network can run in real time and is suitable for thin, long plant parts with various shapes. The next phase is pruning point detection, which is also the main content of this paper. In this phase, our research plant is the sweet pepper. We will detect the lowest leaf and its pruning point in 3D space in our research context.

A sweet pepper dataset was created with more than 4000 images. We labeled plant parts such as stems, leaves, and petioles. Figure 1 shows an example of labeling images of sweet peppers. The left image is the RGB image of a sweet pepper. The right image is the labeled image. It consists of four objects: leaves, petioles, stems, and others that we do not concern ourselves with. Each object is encoded in a different color. This type of image is called a semantic image. We took images of sweet pepper on green farms and in the lab. The camera was handheld and moved around the plants. One image should have a stem and other parts of the plant.

An automated pruning system is complex because it includes a mechanical component for robot design and control and a software component for the robot’s visual interface. It decides where to cut. Our research focuses on this part. He et al. [8] reviewed automation in apple tree pruning. The most technical challenges are tree structure identification, cutting location detection, branch diameter measurement, and length measurement. A 3D reconstruction must be done to get this information. In our research, farmers cut the petioles and these pruning points close to the stem to remove sweet pepper leaves. Hence, our system should detect these pruning points through images. The most important point is the lowest leaf of the plant because it is the starting point for pruning. We will focus on recognizing the sweet pepper’s lowest petiole. Figure 2 shows a potential pruning point in a 2D semantic image. However, the lowest petiole of an image does not mean the lowest petiole of the plant. One image describes a piece of the plant from the camera’s viewpoint. The complete plant information could be found if the camera is moved around the plant and takes images. The solution to combining these images is to reconstruct 3D objects from these images. These 3D objects can represent natural plants in a 3D coordinate system. Therefore, 3D reconstruction is an important part of pruning point detection.

During the development of computer vision, there were many improvements in 3D reconstruction. Many algorithms are proposed to archive this information, such as SIFT [9], SURF [10], FAST [11], and ORB [12], which are feature point extract algorithms. This helps us match objects with feature points in a pair of images, create a space map, and estimate the camera position. Other common algorithms are bundle adjustment [13], which is used to optimize frame poses, and ICP [14], an algorithm that could be used to register adjacent frames (combine the adjacent point clouds).

Early on, it often used 2D images as input. Many proposed systems frequently mention the structure from motion (SFM) technique [15]. It includes three main functions: feature extraction, camera motion estimation, and 3D structure recovery. Applying this technique, Yu et al. [16] presented a method by which a depth map can be generated from a sequence of 2D images. However, there is high uncertainty in the 3D reconstruction. It needs many images to decrease this uncertainty. It could slow down the whole system and make it impractical. Nowadays, the cheap depth sensor is quite popular, and 3D image construction for a 3D image is easy. Our application needs an immense view of the plant. Many RGB-D images should be combined. Henry et al. [17] introduced the RGB-D Mapping system, which can create a 3D dense map from an RGB-D camera based on the ICP algorithm. Although it had encouraging results, it cannot run in real-time. Another system presented by Wang et al. [18] used visual and geometry features combined with SFM (Structure from Motion). Its accuracy is high, but it cannot be used in real-time.

At the same time, many visual SLAM systems that perform simultaneous localization and mapping by the camera-based sensor were introduced [19]. These systems also use the above techniques but focus on the camera’s location and run in real-time. At first, monocular cameras were used with MonoSLAM [20,21]. In recent years, low-cost and high-quality RGB-D cameras have been popular, and many RGB-D visual SLAM systems were born and got much attention. Their structures are simple and take up fewer system resources. KinectFusion was proposed in 2011, which used the GPU and created a 3D map in real-time [22]. Later, RTAB-MAP was introduced in 2014. It had a loop detection function based on the bag of words (Bow) [23]. Its latest version supports many camera types. Then the first ORB-SLAM was introduced in 2015 [24]. The ORB-SLAM family [24,25,26] is one of the visual SLAM systems that is quite popular because it is open-source and supports many types of cameras. It improved the ORB algorithm to extract features so that it can run in real-time on low-resource computers and does not use GPU. Although they have the shortcoming that they do not create a dense map, they are suitable for our system. We need an application that can run in real-time with low resources (memory and energy) and detect the camera position. A dense map of complete meaning is not necessary in this case. We only need to make a dense map or 3D point cloud that includes the sweet pepper plants. On the other hand, the 3D surface or 3D pattern reconstruction is not useful. The positions of plant parts in 3D space are the most important. Therefore, we proposed a method to create a 3D point cloud of sweet pepper plants from the output of ORB-SLAM3 [26].

Our research aims to detect the pruning point of the lowest petioles of the sweet pepper plants. It means that after creating a 3D point cloud of a sweet pepper plant, the lowest petiole and the stem of that plant should be detected in the 3D point cloud. Then the pruning point should be identified on that petiole. However, recognizing objects and calculating their positions in a 3D point cloud take much more time than that in a 2D image. Hence, we proposed a method to create a point cloud that includes all known parts, such as stems, leaves, petioles, and potential pruning points detected in 2D images. These parts are encoded with different colors. This point cloud is called a semantic point cloud.

Petioles are soft and small; it is not necessary to identify the diameters, but it requires accurate pruning positions. In many studies relating to plant pruning, lidar cameras or laser scanners were used to detect branches [27,28]. Therefore, the Intel RealSense LiDAR camera L-515, which can produce high-quality RGB-D images, is chosen to capture images. These images are passed to ORB-SLAM3 to get image poses. They are also passed through a semantic segmentation neural network to recognize plant parts and then detect the potential pruning points. After that, 3D semantic point clouds are created. Here, we propose a module to detect the pruning points in a 3D semantic point cloud. The PCL library [29] stores and visualizes the point clouds.

In summary, our contribution is to propose a system that detects the pruning point of the lowest sweet pepper leaf in 3D space.

## 2. Related Works

### 2.1. Semantic Segmentation Neural Network

As mentioned in the introduction, sweet pepper petioles and other parts were detected in 2D images before building the plant’s 3D semantic point cloud. This task cannot be performed by the neural network for image classification or object detection in a bounding box. We need to know which pixels belong to which objects. Therefore, the semantic segmentation neural network was chosen.

Our previous research focused on recognizing tomato plant parts by a proposed neural network [7]. This neural network can explore the depth information from RGB-D images to improve the result. Furthermore, it has a small number of parameters and runs in real-time, which is essential in our system. Its performance has been proven in previous research.

There are similarities between sweet pepper and tomato plants. We used the same neural network structure and made a new dataset to train the model. RGB-D images are taken at many green farms, from seedlings to mature trees, to make our dataset more general and the model more accurate. This neural network structure is different from the other neural networks. It has two stages, so it leads to two-phase training. The model was trained with more than 4000 images and 50 epochs for each phase. The neural network was written in Python 3.9 and used PyTorch 1.8. It was trained and tested on a computer with a GPU from Nvidia, the GeForce RTX 3090, and CUDA version 11.2. The best IOU model reached was 74.86 percent. Figure 3 presents a result of the model in predicting sweet pepper plant parts.

### 2.2. ORB-SLAM3

Three-dimensional reconstruction is an essential step in finding the relative cutting position for the robot, which is represented by the camera. Many systems can perform this task, but they take a lot of time. It requires an application that can run in real-time and take little computer resources because other modules, such as the neural network, must use many resources. Therefore, the ORB-SLAM series is suitable for our system. The ORB-SLAM family is easy to install, only uses a CPU, and can work well with ROS (Robot Operating System) [30]. Additionally, it is open-source software. Its source code can be modified and updated with much support from the community.

There are many updates from the first version of ORB-SLAM [24] to the newest version, ORB-SLAM3 [26], but they have the same structure summarized in Figure 4. They all use ORB to extract features. While ORB-SLAM accepts monocular cameras, ORB-SLAM3 can work with monocular, stereo, and RGB-D cameras using pin-hole or fisheye lens models. In addition, ORB-SLAM3 can perform visual, visual-inertial, and multi-map SLAM. According to their publication, the performance of ORB-SLAM3 is better than many other visual SLAM systems. Finally, it can run on Ubuntu 20.4 and use the new version of related libraries, which is so convenient for us to continue developing an application based on it.

The output of ORB-SLAM3 is the camera’s position at the moment it gets image input. This position can be updated after detecting a loop. ORB-SLAM3 does not create a dense map. Therefore, if sweet pepper exists in the input images, a 3D point cloud should be created by another module based on the frame poses. This process happens when the robot moves along the field of sweet pepper plants.

### 2.3. ICP Algorithm

Most visual SLAM faces cumulative drift problems, which can be solved by loop detection. However, the camera will only go one way in our system. It cannot make a loop. The cutting position must be found online during its movement. The frame pose should be refined when creating a sweet pepper 3D point cloud. Therefore, the ICP algorithm is used to stick the point cloud built from each frame together and eliminate the drift problem.

The algorithm’s main idea is to minimize the mean-square distance metric over six degrees of freedom by testing a set of rotations and translations. This algorithm is often used to align a pair of point clouds. Especially when these point clouds come from two close frames, they share many of the same objects. In this case, the ICP algorithm becomes very efficient and performs its task quickly. However, the speed also depends on the number of points. Our frame size is 640 × 480, so the number of points could be numerous. Therefore, the ICP algorithm is applied to semantic point clouds consisting of plant part points. This saves much time.

## 3. Proposed System

### 3.1. Detect Pruning Points and Pruning Regions in 2D Semantic Images

Sweet pepper petioles are soft, and they have small diameters. They should be cut at a position near the stem, such as in Figure 2. The lowest petiole should be detected and removed in the context of this study. However, this method is designed to identify all potential pruning points on all petioles because the strategies for pruning can change in the future. Our context is one proposed case for evaluating the method.

We proposed a method to detect the pruning points in 2D images in four steps after they have resulted from the semantic segmentation neural network. This result is a semantic image including sweet pepper plant parts. The images are captured with a resolution of 640 × 480, so a semantic image is a matrix (640 × 480), and each pixel is a matrix item having value $a_{ij}$ is as below:(1)$$a_{ij}=\left\{\begin{array}{l} 1,\;\;a_{ij}\;belongs\;to\;stems\\ 2,\;\;a_{ij}\;belongs\;to\;petioles\\ 3,\;\;a_{ij}\;belongs\;to\;leaves\\ 4,\;\;a_{ij}\;belongs\;to\;fruit\\ 0,\;\;a_{ij}\;belongs\;to\;others \end{array}\right.$$

Figure 5 describes these steps and their results. Firstly, stems and petioles are separated into two separate layers, the S layer and the P layer. Each layer is a matrix (640 × 480), and its items represent pixels in a layer. S layer and P layer have values $s_{ij}$ and $p_{ij}$ as below:(2)$$s_{ij}=\left\{\begin{array}{c} 1,a_{ij}=1\\ 0,a_{ij}\neq 1 \end{array}\right.,\;p_{ij}=\left\{\begin{array}{c} 1,a_{ij}=2\\ 0,a_{ij}\neq 2 \end{array}\right.$$

The predicted results always have many uncertainties. The OpenCV-connected component function is used to remove noise. Next, the petioles and stems are enlarged by a 2D convolution operation with a kernel matrix (7 × 7) with all item values of 1. Therefore, each item of the enlarged layer l[i,j] are calculated as in Equation (3):(3)$$l\left[i,j\right]=\left\{\begin{array}{c} 1,\;\sum_{m=1}^{7}\sum_{n=1}^{7}x\left[i-m,j-n\right]>0\\ 0,\;\sum_{m=1}^{7}\sum_{n=1}^{7}x[i-m,j-n]=0 \end{array}\right.$$where *x* is the stem layer or the petiole layer. The intersection of the two layers is found by adding these two enlarged layers and selecting items having values of 2, as in Equation (4):(4)$$it[i,j]=\left\{\begin{array}{c} 0,\;s\left[i,j\right]+p\left[i,j\right]\neq 2\\ 1,\;s\left[i,j\right]+p\left[i,j\right]=2 \end{array}\right.$$

These intersection regions prove that the petioles connect with the stems and are the petioles’ root points. The pruning points should be near the intersection regions and belong to the petioles. Therefore, in the next step, the intersection layer is enlarged, and its overlapping regions with the original petiole layer are detected. Here, the second intersection layer is created. The points of these intersection regions meet the requirement of pruning points and are called pruning regions. The center points of these regions are chosen as the pruning points. The way to enlarge the layer and find the intersection layer is similar to Equations (3) and (4). Then, these pruning regions are integrated into the original semantic images to create a 3D semantic point cloud.

### 3.2. Create a 3D Semantic Point Cloud

Creating a 3D point cloud is the same as putting each pixel or group of pixels in RGB images as a point in a 3D coordinate system. With the intrinsic parameters of a depth camera, such as the scaling factor, focal length, and center *x* and center *y*, the coordinate (*x*, *y*, *z*) value can be calculated in 3D space as follows:(5)$$z=\frac{depth\left(u,v\right)}{scaling\_factor}\;x=\left(u-centerX\right)*\frac{z}{focal\_length}\;y=\left(v-centerY\right)*\frac{z}{focal\_length}$$where *u* and *v* are the column and row values of a pixel in 2D images, and depth (*u*, *v*) returns the distance value from the camera to the pixel saved in the depth image.

In RGB-D images, each RGB image has a corresponding depth image. When the RGB image is converted to a semantic image, this semantic image shares the same depth image. Therefore, a 3D semantic point cloud can be created from this semantic image and its corresponding depth image.

Figure 6 shows two examples of creating point clouds from a sweet pepper plant displayed by the PCL library at a resolution of 0.001 m. The left image of row (a) is a 3D normal point cloud created by the RGB-D image, while the left image of row (b) is a 3D semantic point cloud created by the semantic image and its depth image. The camera was set up to be about 1.0 m from the plants, so only points with a depth value of fewer than 1.5 m are selected. The black pixels are not plant parts. They are removed when creating the point cloud. The blue regions are pruning regions detected by our proposed method. The semantic point cloud has 38,138 points. This number is much smaller than the number of points in the normal point cloud, which is 107,316. This saves time when applying the ICP algorithm to create these point clouds.

### 3.3. Detect Pruning Points in the 3D Semantic Point Cloud

Each point in a 3D semantic point cloud has a color. This color has its own meaning. In our context, blue points represent pruning regions, and the pruning points in 3D space must be in these areas as well. However, there are many blue regions in the semantic point cloud. The lowest pruning region and its position should be detected. Therefore, a four-step method to find the lowest pruning points in 3D space is proposed as in Figure 7.

Firstly, the pruning point cloud is extracted by selecting only blue points from the 3D semantic point cloud. A 3D point cloud is created by many RGB-D images taken from different viewpoints. So that a pruning region in a 3D point cloud can be created from diverse groups of points from different viewpoints, and these groups can be unconnected to each other. Therefore, in step 2, the closet points are inserted into each group to connect them. Then, each group of points is a pruning region of a petiole. Next, each group is separated, and small groups seen as noises are removed. The y values in the 3D coordinate system between the remaining groups are compared. The group with the smallest y value is the lowest pruning region, belonging to the lowest petiole. Finally, the center point of this group is the final pruning point. The red point in the white circle in the right image of Figure 7 is the pruning point of the lowest petiole.

The PCL library can store points in a Kd-tree (k-dimensional tree) data structure. It provides a function to find points having a distance of less than a value. A method is proposed to extract each group of pruning region points by applying this function, as in Figure 8. Firstly, a random point, P, is selected in the point cloud. This will be the first point of a group. Then all other points having a distance to this point less than a threshold t are detected. These points are put into a stack. Next, point P is removed from the point cloud. Then, if the stack size is greater than 0, a point is popped and put into the group, and then a loop is made as in Figure 8. When the stack is empty, one group has been found completely. A new round is performed until there are no more points in the point cloud. The value of t depends on the point cloud’s resolution value, r. We choose t = 1.5 × r.

### 3.4. The Entire Pruning Points Detection System

The system consists of five main modules: create semantic images by semantic segmentation neural networks and detect the pruning regions, find camera poses by ORB-SLAM3, create semantic point clouds, register different point clouds by ICP, and detect pruning points in the 3D semantic point clouds. Figure 9 shows the structure of this system. Section 3.2 shows how to create a semantic point cloud from a semantic and depth image. However, the semantic point cloud should be made from many images. It means that many semantic point clouds should be aligned. Therefore, we proposed a method using ORB-SLAM3 to get the camera poses to construct 3D point clouds in the same coordinate system. The ICP algorithm is applied to refine these point clouds to reduce the drift that has accumulated over time.

After receiving a start signal, RGB-D images were passed through the semantic segmentation neural network to recognize the plant parts. Then, these results go through the pruning region detection module. On the other hand, RGB-D images were passed through ORB-SLAM3 to detect the camera poses. These are parallel threads. Semantic point clouds are created at the end of both threads based on the previous results. This process will be repeated until the system gets the stop signal. The ICP algorithm is applied at each loop to align the last semantic point cloud with the new one. When receiving a stop signal, the creation of a 3D semantic point cloud process stops, and the final semantic point cloud is passed through the 3D pruning point detection module to get the final pruning point position.

## 4. Experiment and Results

### 4.1. Experiment

Our experiments focus on accurate visual results and stable values of pruning point positions. The system is reliable if the pruning points in the point clouds created by different camera trajectories are the same and the difference in pruning position values is small. The time of execution is also estimated to verify if the system can be applied in practice.

The experiments were set up in the lab with sweet pepper plants raised in flower pots. There is some breeze making the plant tremble slightly. The light does not shine directly on the camera. We used one camera, the Intel RealSense L515, to capture RGB-D images. The depth and RGB images are aligned before processing by our application. The camera was placed in three different positions for a sweet pepper plant, as in Figure 10. At each position, the camera was handheld and moved with four different trajectories: up, left, right, and a combination of three of them. The camera was stopped at the starting position.

The application was written as ROS nodes under Ubuntu 20.4. There are three nodes. One node wraps ORB-SLAM 3 to provide camera poses. One node was written in Python to predict plant parts by using a semantic segmentation neural network and detect pruning regions from RGB-D images. The final node was written in C++, creating a 3D semantic point cloud and finding a pruning point in 3D space. The system was installed in a laptop having a CPU of i7–7500U, and a GPU of Nvidia GeForce 940MX.

### 4.2. Results

The average time for creating and aligning a semantic point cloud is 0.725 s. The total time to find a pruning point depends on the length of the camera path before stopping.

We got the 3D semantic point clouds for three different camera view points and four different camera movement directions. Figure 11 shows the twelve semantic point clouds created by the experiments. The point clouds in one row had the same starting and stopping positions. The point clouds in each column had the same camera moving directions. In these point clouds, the green, pink, and purple colors represent the leaves, petioles, and stems, respectively. The light blue dot in each blue point group is the pruning region center point. The pruning point of the lowest petiole was marked by surrounding red points. Although different experiments used different viewpoints, their pruning points in the point clouds were the same. Some point clouds were blurred, but the pruning point positions were still correct. In the experiments, there were some sudden changes in positions and the speed of the camera, which was moved by hand. Additionally, the wind made the position of soft leaves change. These reasons could make the point cloud blurry. So, PCL filter libraries are used to remove outliers and noise points. The most common positions of the plant parts were kept. Thanks to PCL filter libraries, some noise in plant part predictions by the semantic segmentation neural network was also removed.

Table 1 presents the pruning points’ position in the 3D coordinate system in which the camera position is (0,0,0). One unit of the 3D coordinate system represents 1 mm. The average difference is the distance from the pruning point detected on the mixed path to other pruning points with the same stopping positions. These values show that the positions of pruning points with the same stopping position are close. With around 400mm of distance from the camera to the pruning points, a little difference in stopping place could lead to a considerable difference in the pruning points. Furthermore, the camera is moved by a human hand, so the stopping positions could make little difference in our experiments. Therefore, the average difference from 4.1 to 6.2 in Table 1 proved that our system is stable.

### 4.3. Discussion

Our experiments worked on natural sweet pepper plants, and the camera was moved by hand. Estimating the ground-truth value to evaluate the experiment results took much work and was expensive. However, in every hardware system, there are always differences. If it is stable, it can be refined after experiments.

Twelve experiments were conducted on a plant to check the system’s accuracy. Although the camera’s moving path and starting and stopping positions were different, they returned the same results visually. This proved that the semantic point clouds were created correctly. The correct visual results do not mean the right distance or position of the pruning points to the camera. Therefore, the experiments are divided into three groups. Each group has the same camera stopping position. The pruning point positions are detected and compared to each other in a group. If the differences in each group were significant, the distance from the camera to the pruning point was detected incorrectly. However, the experiments showed that this value was small in each group. It proved that the pruning point positions detected by the system are correct and reliable.

The experiment environment was set up with lots of light and breeze to simulate a natural environment. When the robot moves, it could make plants tremble, and the depth information in RGB-D images could change for the same point. It is a challenge, but our system can handle this with acceptable results. The first row of Figure 11 is an example of a trembling plant.

After archiving with the required accuracy, we focus on time consumption. Most of the time is taken by the creating semantic point cloud process. Creating a semantic point cloud of pair RGB-D images in these experiments takes around 0.7 s. It is acceptable because the robot cannot move fast, and images should be taken at a reasonable distance to get more different viewpoints. In addition, if the camera is moved quickly, it could lead to errors when applying the ICP method because the overlap between a pair of point clouds is small. More experiments will be done to detect the maximum moving speed of the camera when the program is applied to a complete robotic system.

There are little differences in time, resources, and accuracy when the images of the plants are taken in the field versus those of the potted plant in the laboratory. The plants in the fields are bigger, but we are interested only in the lower part of the plant for pruning. Hence, if we capture images of the lower part of the plant, the time, resources, and accuracy are similar. We took the RGB-D images at many green farms from seedling to mature age to make our dataset more general and the model more accurate.

Similar to many other SLAM systems, drift is a big problem. If the camera moves along a curving path, it could lead to errors in creating a semantic point cloud. The camera moving in a straight line is a good way to decrease this problem. In addition, our point cloud is made from many viewpoints. One petiole may not be seen in one depth image, but it can be seen in other depth images. It decreased the occlusion problem and the impact of the light-to-depth images. We are working on a prototype of the system. This research took part, which is taking images and returning the cutting position. We will do more research on controlling robots before experimenting on green farms.

## 5. Conclusions

We proposed a system to detect sweet pepper leaf pruning points based on a 3D point cloud and a semantic segmentation neural network. This system can run in a real-time and realistic environment. Our experiments proved that the proposed system is stable and reliable, with correct visual results and a distance of pruning points from the camera.

In addition, in an attempt to achieve a complete sweet pepper pruning system, this research addressed the important task of providing cutting positions. The proposed system can be applied to the robot’s arm or manipulator with a scissors end effector. In the future, we will continue to study an entire robotic system. This research can be extended to perform sweet pepper shot pruning, width, and height, or leaf area detection based on 3D semantic point clouds, and it can be a good reference for other plant pruning systems. Furthermore, a 3D semantic point cloud can be used for object detection in 3D space. It would use fewer resources than training a neural network to detect an object in the original 3D point cloud.

## Figures and Tables

**Figure 1 sensors-23-04040-f001:**
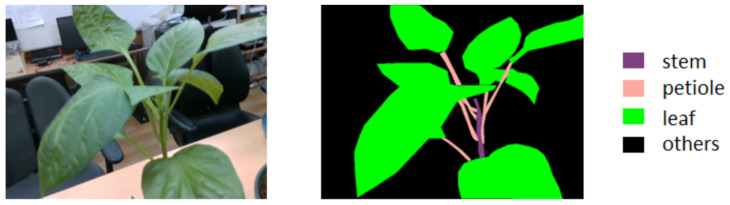
An example of sweet pepper labeling.

**Figure 2 sensors-23-04040-f002:**
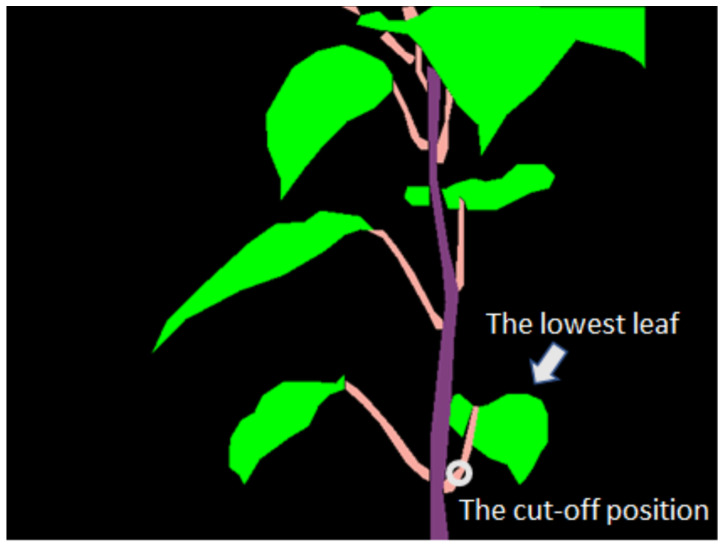
A potential pruning point in a 2D semantic image.

**Figure 3 sensors-23-04040-f003:**
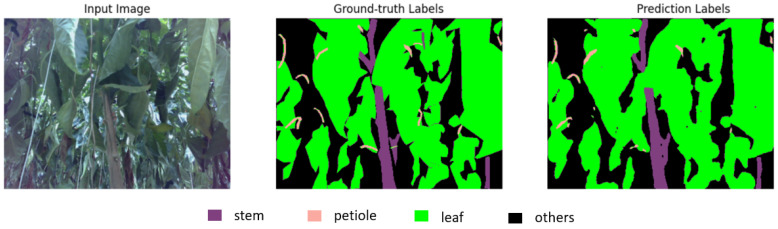
The prediction result of the model on the sweet pepper plant.

**Figure 4 sensors-23-04040-f004:**
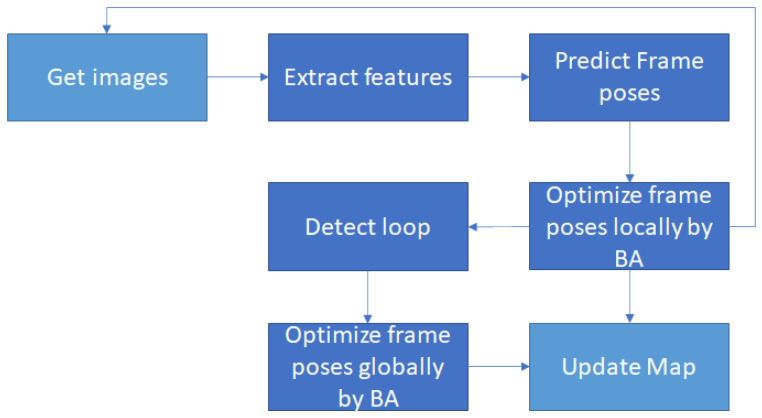
ORB-SLAM series structure.

**Figure 5 sensors-23-04040-f005:**
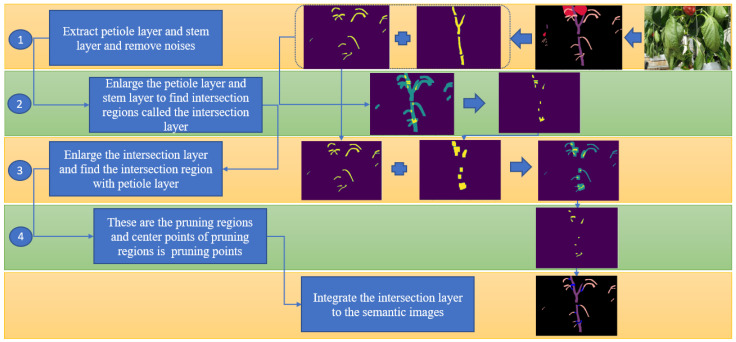
Four steps to detect pruning regions and pruning points in 2D semantic images.

**Figure 6 sensors-23-04040-f006:**
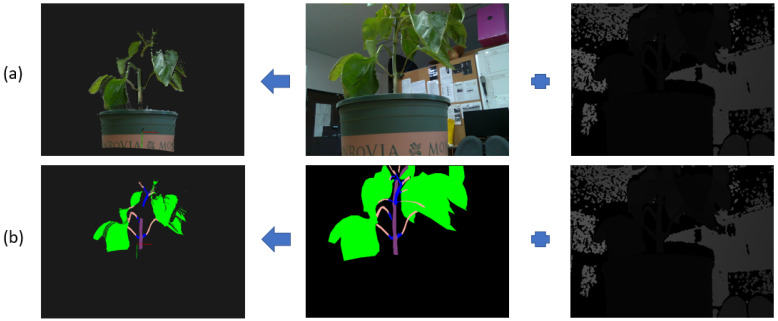
Two point clouds created from a sweet pepper plant. (**a**) 3D normal point cloud created from an RGB and depth image. (**b**) 3D semantic point cloud created from semantic and depth images. The red and green lines represent X and Y axes in a 3D coordinate system.

**Figure 7 sensors-23-04040-f007:**
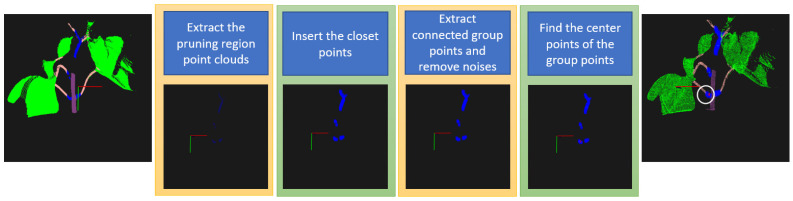
Four steps to detect the lowest pruning point in 3D space, and the red point is the final pruning point in the white circle. The red and green lines represent X and Y axes in a 3D coordinate system.

**Figure 8 sensors-23-04040-f008:**
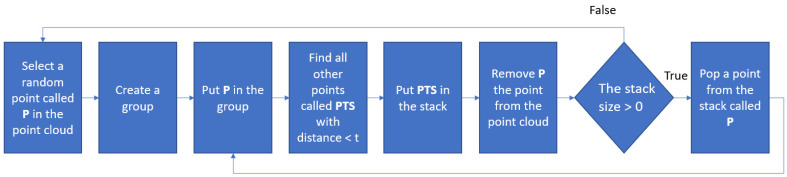
The diagram of extracting pruning group points.

**Figure 9 sensors-23-04040-f009:**
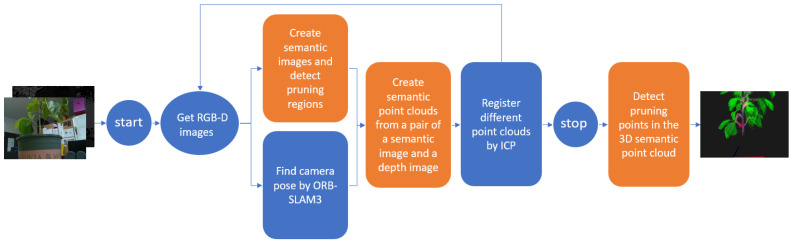
Structure of the sweet pepper leaf pruning point detection system. The orange rectangles are our proposed modules. The red point in the final semantic point cloud is the pruning point of the lowest petiole.

**Figure 10 sensors-23-04040-f010:**
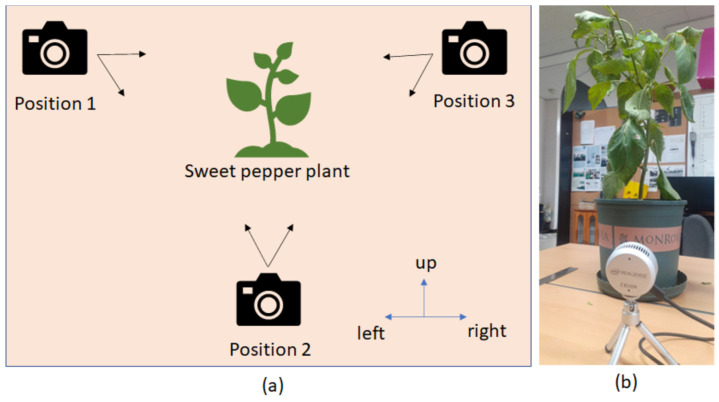
Setting up the experiments. (**a**) shows camera positions, and (**b**) is a camera position example of experiments. The black arrows are camera viewpoints. The blue arrows are moving camera directions.

**Figure 11 sensors-23-04040-f011:**
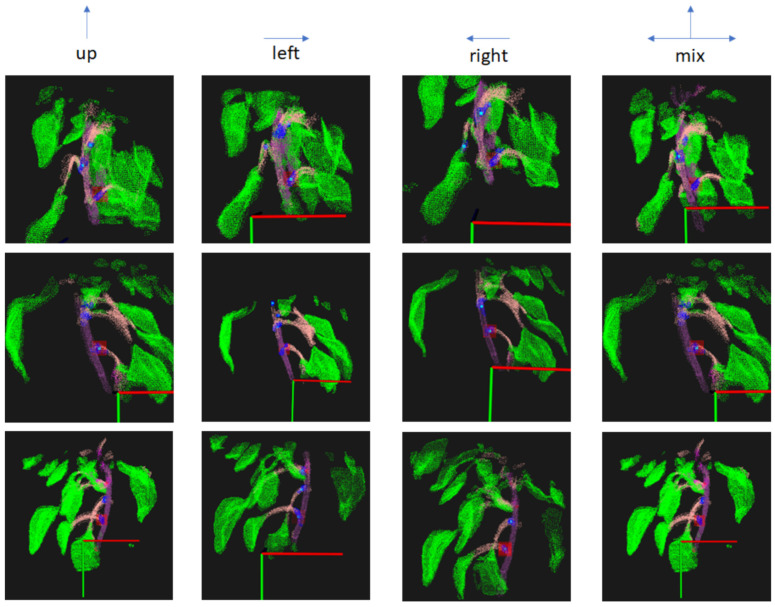
The images of twelve-point clouds were created through experiments. The pruning points were a light blue dot surrounded by a red area to be recognized easily by human eyes. The red, green, and black lines represent X, Y, and Z axes in a 3D coordinate system.

**Table 1 sensors-23-04040-t001:** The 3D coordinate value of pruning points was detected from twelve camera moving paths.

Camera	Up (x, y, z)	Left (x, y, z)	Right (x, y, z)	Mix Path (x, y, z)	Average Difference (mm)
Position 1	(15,−49,400)	(12,−45,401)	(22,−47,393)	(16,−48,401)	4.19
Position 2	(−17,−75,388)	(−17,−76,391)	(−23,−75,388)	(−15,−80,394)	6.15
Position 3	(50,−53,374)	(46,−54,374)	(50,−46,372)	(46,−51,376)	4.01

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
