# Peer review of "Pruning Points Detection of Sweet Pepper Plants Using 3D Point Clouds and Semantic Segmentation Neural Network"

_sensors, 2023, doi:10.3390/s23084040_

Round 1
Reviewer 1 Report
In this work, the authors proposed a system to detect sweet pepper leaf pruning points based on 3D point cloud and semantic segmentation neural network. The reported method is potentially useful for practical application. I think this manuscript can be accepted after some revisions. Here are some specific questions/comments.
1. The authors proposed the 3D coordinate system, where does it have the greatest advantage over other vision systems of similar type.
2. The 3D coordinate system can only detect the pruning point of the lowest pepper leaf. Is this a defect of it?
3. I think the authors should have collected more images from different growth of sweet peppers for machine learning. More statistically valid data should be provided.
Author Response
Thank you very much for your comments.
Please, find the attached file of the answers.

Reviewer 2 Report
The manuscript entitled "Design of Fuzzy-PD Controller for Magnetic Field Guided Differential Drive Mobile Robot" has very good and novel results and findings which will be interesting for readers. The authors focus on image analysis system for defining the plant parts for harvesting. The abstract, Introduction as well as research method and results are well designed. Only there are some typos and English grammatical mistakes that should be considered and corrected before acceptance for publication. Also, it is recommended that authors add more references to the manuscript.Author Response
Thank you very much for your comments.
Please, find the attached file of the answers.

Reviewer 3 Report
I believe that the authors selected and combined the correct methodologies and tools for the applied research, namely: Semantic segmentation neural network, ORB-286 SLAM3, ICP algorithm. They conveniently cited the difficulties and limitations of each of these for the experiment and research they are conducting which requires real-time processing.
The three proposed modules are interesting and allow dealing with the required information with a smaller amount of computer resources. Thus, the creation of semantic point clouds from RGB-D images from multiple cameras (or one camera moving around the plant), the detection of plant parts (sweet bell pepper) and the registration of these with their labels, allows them to save time and computational capacity; and allows them to identify the pruning point according to plant characteristics with good accuracy.
I find that the article is in line with the objectives of the journal and that it will be useful for other research related to the many activities that need to be automated in agriculture and the great variety of crops. Therefore, I consider that the detail with which the methodology and results are presented is very relevant. However, I would have liked to know a little more about the differences in time, resources and accuracy when the images of the plants are taken in the field vs. those of the potted plant in the laboratory.
Author Response

(The authors gave the same response as above.)

Reviewer 4 Report
1) The authors applied computer vision and space mapping combined with machine learning in an attempt to improve the guidance system of a robot for pruning and picking sweet peppers. Their goal is to reduce handling by hand in order to limit the spread of plant disease. 2) Computer vision, space mapping and machine learning have been applied to robots so far. The novelty is in their combined application. 3) The contribution is in the application to a new area (agriculture). 4) I believe that the methodology was adequate and the controls appropriate. 5) The conclusions are consistent with the research hypotesis and presented evidence. The authors have not described to what kind of robot they intend to apply the control system though. This should be expanded further. Also the conclusion should be expanded to cover other applications outside agriculture. I was unsure about the article fitting the scope of Sensors as there was no mention of the hardware it was supposed to run in the real world application. I would still recommend publishing as I consider the article scientifically sound and the research complete and well documented. 6) The references are adequate. However the number of references related to the area of engineering and computer science should be increased. Also, a lot of references are conference papers. The balance should be shifted to favour journal articles. 7) The tables are of good quality. The figures are good and in high resolution. Figures 1,2,3,6,7 and 11 should be enlarged for clarity if possible.Author Response
Thank you very much for your comments.
Please, find the attached file of the answers.
